# Use of NEedle Versus suRFACE Recording Electrodes for Detection of Intraoperative Motor Warnings: A Non-Inferiority Trial. The NERFACE Study Part II

**DOI:** 10.3390/jcm12051753

**Published:** 2023-02-22

**Authors:** Sebastiaan E. Dulfer, Maria C. Gadella, Katalin Tamási, Anthony R. Absalom, Fiete Lange, Carola H. M. Scholtens-Henzen, Christopher Faber, Frits H. Wapstra, Rob J. M. Groen, Marko M. Sahinovic, Sedat Ulkatan, Gea Drost

**Affiliations:** 1Department of Neurosurgery, University Medical Centre Groningen, University of Groningen, 9713 GZ Groningen, The Netherlands; 2Department of Anaesthesiology, University Medical Centre Groningen, University of Groningen, 9713 GZ Groningen, The Netherlands; 3Department of Epidemiology, University Medical Centre Groningen, University of Groningen, 9713 GZ Groningen, The Netherlands; 4Department of Neurology, University Medical Centre Groningen, University of Groningen, 9713 GZ Groningen, The Netherlands; 5Department of Orthopaedics, University Medical Centre Groningen, University of Groningen, 9713 GZ Groningen, The Netherlands; 6Department of Intraoperative Neurophysiology Mount Sinai West Hospital, New York, NY 10029, USA

**Keywords:** muscle-recorded transcranial electrical stimulation motor evoked potential, recording electrodes, warning criteria, neurological outcome

## Abstract

In the NERFACE study part I, the characteristics of muscle transcranial electrical stimulation motor evoked potentials (mTc-MEPs) recorded from the tibialis anterior (TA) muscles with surface and subcutaneous needle electrodes were compared. The aim of this study (NERFACE part II) was to investigate whether the use of surface electrodes was non-inferior to the use of subcutaneous needle electrodes in detecting mTc-MEP warnings during spinal cord monitoring. mTc-MEPs were simultaneously recorded from TA muscles with surface and subcutaneous needle electrodes. Monitoring outcomes (no warning, reversible warning, irreversible warning, complete loss of mTc-MEP amplitude) and neurological outcomes (no, transient, or permanent new motor deficits) were collected. The non-inferiority margin was 5%. In total, 210 (86.8%) out of 242 consecutive patients were included. There was a perfect agreement between both recording electrode types for the detection of mTc-MEP warnings. For both electrode types, the proportion of patients with a warning was 0.12 (25/210) (difference, 0.0% (one-sided 95% CI, 0.014)), indicating non-inferiority of the surface electrode. Moreover, reversible warnings for both electrode types were never followed by permanent new motor deficits, whereas among the 10 patients with irreversible warnings or complete loss of amplitude, more than half developed transient or permanent new motor deficits. In conclusion, the use of surface electrodes was non-inferior to the use of subcutaneous needle electrodes for the detection of mTc-MEP warnings recorded over the TA muscles.

## 1. Introduction

Different stimulation settings, techniques and methods are used for intraoperative neuromonitoring (IONM) with muscle-recorded transcranial electrical stimulation motor evoked potentials (mTc-MEPs) [1]. The most commonly used method is the amplitude reduction method, which can be performed with either submaximal or supramaximal stimulation [1,2]. For submaximal stimulation, the voltage is usually 20–30% above the lowest stimulation intensity required to generate a motor threshold [2]. During supramaximal stimulation, the stimulation intensity is increased beyond the stimulation intensity necessary to evoke the maximum mTc-MEP amplitude [1,3].

IONM aims to prevent intraoperative neurological injury [4,5]. If signs of impending spinal cord injury are detected early, the operating theatre team can take measures to prevent the injury from becoming permanent. With the amplitude reduction method, the criterion for impending injury is a percentage decline from baseline in the mTc-MEP amplitude. Different relative amplitude decreases have been proposed as warning criteria for the amplitude reduction method [4,6,7], but there is no consensus on the optimal relative decline. Depending on the type of surgery, warning criteria of mTc-MEPs range from a 50% to 100% decrease in amplitude compared to the baseline. With supramaximal stimulation—which is associated with lower inter-trial variability—a warning criterion of ≥80% is associated with a positive predictive value of only 0.6 [1]. Therefore, a 50% warning criterion associated with an even lower positive predictive value and higher false positive rate might be too sensitive.

Different types of recording electrodes have been proposed for recording mTc-MEPs [4]. A distinction can be made between extramuscular recording electrodes, consisting of surface and subcutaneous needle electrodes, and intramuscular recording electrodes, consisting of needle and hookwire electrodes [4,8,9].

In part I of the NERFACE study, mTc-MEP monitoring characteristics were compared between subcutaneous needle and surface recording electrodes [10]. We concluded that both recording electrodes were acceptable for mTc-MEPs for spinal cord monitoring. It is however not known whether mTc-MEP recordings from surface electrodes are of equivalent utility to mTc-MEPs recorded from subcutaneous needles for detection of mTc-MEP warnings during spinal cord monitoring.

Therefore, this study aims to investigate if the use of surface recording electrodes is non-inferior to the use of subcutaneous needle recording electrodes for the detection of mTc-MEP warnings during spinal cord monitoring. Since Kim et al. reported that recordings from the TA were better predictors of new postoperative motor deficits than recordings from the abductor hallucis muscles, we recorded mTc-MEPs from the left and right TA muscles in this study [11].

## 2. Materials and Methods

### 2.1. Study Design

This was a prospective observational study. Since the data were collected during routine clinical care, the hospital ethical committee waived the requirement for full ethical committee review following the terms of the Dutch Act on Medical Research on Human Subjects (Wet Medisch-Wetenschappelijk Onderzoek, or “WMO”). The study was, however, registered and approved by a non-WMO study evaluation committee which deemed informed consent unnecessary.

### 2.2. Patients

Patients were included from March 2019 until January 2022. The inclusion criteria consisted of (1) ≥12 years old, (2) patients underwent spinal cord monitoring with mTc-MEPs, (3) ≥10 mTc-MEP measurements were performed, and (4) mTc-MEPs were elicitable in at least one TA muscle (left or right).

### 2.3. Objectives

The primary objective of this study was to investigate if the use of surface recording electrodes was non-inferior to the use of subcutaneous needle recording electrodes in detecting mTc-MEP warnings for the left and right TA muscles during spinal cord monitoring. The secondary objective was to investigate how well mTc-MEP warnings, recorded with either surface recording electrodes or subcutaneous needle recording electrodes, predict postoperative neurological injury of the left and right TA muscle during spinal cord monitoring.

### 2.4. Muscle-Recorded Transcranial Electrical Stimulation Motor Evoked Potentials

The mTc-MEP monitoring method and settings were described in the NERFACE study part I [10]. Surface and subcutaneous needle electrodes were placed at the TA muscles. A detailed description of the placement of recording electrodes has been described in the NERFACE study part I [10].

An mTc-MEP warning was defined as a reproducible significant deterioration or complete loss of mTc-MEP amplitude of the left or right TA muscles, with other possible causes of deterioration or loss (i.e., technical issues, low blood pressure, increased propofol concentration) ruled out. If a warning occurred, it was communicated to the surgeon. The warning criteria depended on the type of surgery (Table 1) and were at either ≥50% or ≥80% deterioration of mTc-MEP amplitude [7]. If an mTc-MEP warning occurred only at one type of recording electrode, the warning was communicated to the surgeon. Different measures were employed to reverse the amplitude decline if a warning occurred. Measures included (1) surgical pause, (2) decreased propofol target concentration, (3) increased blood pressure, (4) irrigation of the spinal cord, (5) removal of traction/screws/hooks/sheaths, (6) autologous blood transfusion (only in the presence of anaemia) using blood recovered with the Xtra Cell Saver device (LivaNova, London, United Kingdom).

For the primary objective, mTc-MEP monitoring results were classified as “no warning” or “warning”. For the secondary objective, mTc-MEP monitoring results were classified as follows: (1) no warning, (2) reversible deterioration or loss, (3) irreversible deterioration, or (4) irreversible loss. Postoperative neurological outcomes were classified as: (1) no new postoperative motor deficits, (2) transient new postoperative deficits, or (3) permanent new motor deficits. Muscle strength was determined using the Medical Research Council (MRC) scale [12]. The MRC grades range from 0 (no visible contraction) to 5 (active movement against full resistance).

### 2.5. Anaesthesia

Anaesthetic management was described in NERFACE study part I [10]. In short, for general anaesthesia, total intravenous anaesthesia (TIVA) using propofol with either sufentanil or remifentanil was used during all surgeries. In 110 (52.4%) patients, esketamine was administered as an analgesic adjunct.

### 2.6. Data Collection

mTc-MEP warnings recorded at the left and right TA muscles were collected from the clinical neurophysiologist IONM reports. mTc-MEP signals were exported from the NIM-Eclipse E4 IONM system (Medtronic BV, the Netherlands) after which the mTc-MEP amplitudes were calculated and collected using software routines written in Python (version 3.7.1). The consecutive mTc-MEP amplitudes were plotted per patient, muscle and type of recording electrode for visual inspection to objectify the warnings from the clinical neurophysiologist IONM reports. Pre- and postoperative MRC grades from the left and right TA were collected retrospectively from the electronic patient records.

### 2.7. Sample Size Calculation

The sample size calculation was performed using PASS V.15 (NCSS, Kaysville, Utah, USA). The subcutaneous needle recording electrode was considered the gold standard for the sample size calculation. Based on an interim analysis of the first 53 patients, in which six patients had a warning detected by both surface and subcutaneous needle electrodes, we assumed a warning proportion of 0.113 (6/53). A sample size of 210 subjects achieved 97.4% power at a 5.0% significance level using a one-sided equivalence test of correlated proportions when the standard proportion, the proportion of the warnings detected by the subcutaneous needle electrodes was 0.113. Since there was a perfect agreement in the interim analysis between the recording electrode types for the detection of an mTc-MEP warning, the maximum allowable difference between the correlated proportions that still results in equivalence (the range of equivalence) was specified at 5.0% and the actual difference of the proportions was 0.0. The observed final warning proportion was 0.119 (25/210), which was remarkably similar to the proportion from our interim analysis (proportion = 0.113). Therefore, we considered the interim sample size calculation valid for our final analysis.

### 2.8. Statistical Analysis

All analyses were performed in R Software version 4.0.5. (The R Foundation for Statistical Computing). Normally distributed variables were summarized using their mean and SD, while non-normally distributed as the median and interquartile range (IQR). Non-inferiority of the primary outcome was tested with the exact McNemar (paired binary) test with difference in proportions (mcnemarExactDP function in R from the exact 2 × 2 package [13]) at a one-sided 95% CI with a non-inferiority margin of 5.0%. The fragility index—the minimum number of patients with a warning detected by the subcutaneous needle recording electrode whose status (warning or no warning) must change to convert a statistically significant result to non-significant—was also calculated as a measure of robustness [14].

For the secondary outcome, monitoring results and neurological outcome were reported as frequencies in a 4 by 3 table separately for both the left TA and right TA. The sensitivity, specificity, positive predictive value, and negative predictive value were calculated separately for the left and right TA. To be able to calculate the predictive values, irreversible deterioration and irreversible loss of the mTc-MEP amplitude were defined as a warning. Transient and permanent new postoperative motor deficits were defined as motor deficits. To classify reversible warnings, we applied the method proposed by Skinner et al. [15]. The predictive values were calculated separately, in which the reversible warnings were either (1) excluded from the analysis, (2) defined as true positives, (3) defined as false positives, or (4) defined as true or false positives after the application of Hill’s causality guidelines [16]. Regarding the 4th category, only mTc-MEP warnings that reversed after removal of traction/screws/hooks/sheaths, were considered true positive warnings, similar to method reported by Skinner et al. [15].

Confidence intervals of the predictive values were calculated with the MedCacl diagnostic test evaluation calculator (MedCalc Software version 20.123, MedCalc Software Ltd., Ostend, Belgium).

## 3. Results

### 3.1. Patients

From the 242 consecutive patients, 32 (13.2%) were excluded from this study (Figure 1). Twenty-one patients had fewer than ten mTc-MEP measurements, in seven patients mTc-MEP data could not be exported, in three patients, neither the left and right TA were elicitable, and in one patient, the surface electrodes might have been incorrectly connected to the recording device. Therefore, 210 patients were included in the final analysis. Patient characteristics are shown in Table 2.

### 3.2. Primary Outcome

In 25 (11.9%) out of 210 patients, an mTc-MEP warning was detected from the left and/or right TA (Table 3). There was a perfect agreement between both recording electrode types for the detection of mTc-MEP warnings.

The proportion of patients with a warning detected by the subcutaneous needle electrode was 0.12 (25/210), which was equal to the proportion of patients with a warning detected by the surface recording electrode (difference, 0.0% (one-sided 95% CI, 0.014)). As the point estimate and CI of the difference were 5.0% and 3.6%, respectively, lower than the non-inferiority margin of 5.0%; the non-inferiority of the use of surface recording electrode in detecting warnings was demonstrated (Figure 2). The fragility index—the minimum number of patients with a warning detected by the use of a subcutaneous needle recording electrode whose status (warning or no warning) must change to convert a statistically significant result to non-significant—yielded six patients.

Warnings in both the left and right TA were detected in 16 patients out of 25 patients (64.0%). In five patients (20.0%), there was a warning detected in only the right TA, and in four patients (16.0%), there was a warning detected in only the left TA. Warnings always occurred simultaneously for both recording electrodes, i.e., a significant deterioration or complete loss of the mTc-MEP amplitude of the TA muscles never started earlier or later for one type of recording electrode.

Patient characteristics of the 25 patients with a warning are shown in Table 4.

### 3.3. Secondary Outcome

For the secondary objective, monitoring results were separately compared to the postoperative neurological outcome for the left and right TA (Table 5). Since there was a perfect agreement in monitoring results between surface recording electrodes and subcutaneous needle electrodes, no distinction was made between the recording electrodes for the secondary outcome.

There were no false negatives for the left TA since all 185 patients (100.0%) without warning had no new postoperative motor deficits. For the right TA, one patient (0.5%) out of 186 patients without warning had transient new postoperative motor deficits. In this patient, the preoperative MRC of the right TA was 5 and the postoperative MRC was 4+. After three days, the MRC was 5 again.

Seven patients (58.3%) (patients 16-18, 20-23, Table 4) out of twelve patients with an irreversible deterioration or complete loss of the mTc-MEPs had either transient or permanent new postoperative motor deficits.

For the left TA, ten patients had an irreversible deterioration or complete loss of the mTc-MEP amplitude. One of them (10.0%) had permanent new postoperative motor deficits (patient 23, Table 4; follow-up: 301 days). Four patients (40.0%) had transient new postoperative motor deficits. Five patients (50.0%) had no new postoperative motor deficits of the left TA.

For the right TA, 11 patients had either an irreversible deterioration or complete loss of the mTc-MEP amplitude. Two patients (18.2%) had permanent new postoperative motor deficits (patient 22 and 23, Table 4; follow-up 77 and 301 days, respectively). Five patients (45.5%) had transient new postoperative motor deficits. Four patients (36.4%) had a false positive warning since they had no new postoperative motor deficits of the right TA muscle.

Nine patients had a reversible deterioration or loss of the left TA amplitude. Eight (88.9%) out of the nine patients did not have new postoperative motor deficits involving the left TA. One patient (11.1%) had a transient new postoperative motor deficit involving the left TA. All nine patients (100.0%) who had a reversible deterioration or loss of the right TA amplitude did not have new postoperative motor deficits. In Figure 3 a case example of a reversible warning is shown (patient 19, Table 4).

Predictive values were calculated for the left and right TA muscles and are shown in Table 6. The predictive values were calculated separately, in which the reversible warnings were either (1) excluded from the analysis, (2) defined as true positives, (3) defined as false positives or (4) defined as true or false positives after application of the Hill’s causality guidelines [16]. After application of the Hill’s causality guidelines, for the left TA, two (22.2%) out of the nine patients with a reversible warning were considered true positive (patients 4 and 11, Table 4). For the right TA, five (55.6%) out of the nine patients with a reversible warning were considered true positive (patient 4 and 12–15, Table 4).

## 4. Discussion

This study investigated whether the use of surface recording electrodes was non-inferior to the use of subcutaneous needle electrodes for the detection of mTc-MEP warnings recorded over the TA muscles during spinal cord monitoring. There was a perfect agreement between both recording electrodes for the detection of mTc-MEP warnings. The use of surface electrodes was non-inferior to the use of subcutaneous needle electrodes for the detection of mTc-MEP warnings recorded over the TA muscles. Moreover, reversible warnings were never associated with permanent new motor deficits involving the left and right TA muscles. Among patients with irreversible warnings, transient or permanent new postoperative motor deficits were found in 5 (50.0%) out of 10 patients in the left TA and in 7 (63.6%) out of 11 patients (63.6%) in the right TA.

This is the first study that compares the frequencies and outcomes of mTc-MEP warnings recorded from the surface and subcutaneous needle electrodes over the TA muscles. All warnings were simultaneously detectable in the mTc-MEPs from both the surface and subcutaneous needle electrodes. The use of surface electrodes was non-inferior to the use of subcutaneous needle electrodes for detecting mTc-MEP warnings recorded over the TA muscles. Therefore, recording mTc-MEP amplitudes from the TA muscles with surface electrodes is as reliable as recording mTc-MEP amplitudes with subcutaneous needle electrodes for detecting mTc-MEP warnings during spinal cord monitoring of the TA muscles. To assess the robustness of our results, the fragility index was calculated. The minimum number of patients with a warning detected by the subcutaneous needle electrode whose status statistically changed our significant result to a non-significant result was *n* = 6. Since the higher the fragility index, the more robust the results, and in a systematic review of 40 surgical spine trials, the median fragility index was *n* = 2 (IQR 1, 3), we consider the fragility index of *n* = 6 to support the robustness of our results [17].

All patients who had a reversible warning, did not have permanent new postoperative motor deficits. Four (44.4%) out of the nine patients with reversible warnings underwent an endovascular aneurysm repair. Although, these warnings were due to the deployment of sheaths in the femoral arteries, and therefore caused ischemia of the peripheral nerves, there still was a perfect agreement between both recording electrodes. Reversible mTc-MEP warnings have proven challenging to classify as either true positive or false positive warnings, since the neurological status at the time of the warning is unknown [4]. Therefore, in this study, we applied the method proposed by Skinner et al., in which Hill’s causality guidelines were used [15,16]. Only mTc-MEP warnings that reversed after removal of traction/screws/hooks/sheaths, were considered true positive warnings, similar to the method reported by Skinner et al. This might be an underestimation of true positive mTc-MEP warnings, since one could argue that increasing blood pressure or administering autologous blood transfusion may have prevented neurological deficits caused by ischemia. Therefore, the classification of reversible mTc-MEP warnings remains challenging.

Monitoring mTc-MEPs from the TA muscles, using the amplitude reduction method, resulted in a high sensitivity and specificity regarding the neurological outcome. There was one false negative mTc-MEP outcome for the right TA. However, the difference in motor strength was small (preoperative MRC 5, postoperative MRC 4+), and different physicians performed the neurological examination. Therefore, it is possible that this difference could have been the result of inter-observer variability, and not a true false negative.

Five patients had an irreversible warning without new postoperative motor deficits (patients 6–9 and 24, Table 4). D-waves were measured in four out of these five patients (patients 7–9 and 24, Table 4). In one patient (patient 8, Table 4) there was no D-wave warning and in one patient (patient 24, Table 4) there was a reversible D-wave warning. It has been reported that patients can have significant mTc-MEP deteriorations while D-wave amplitudes remain stable [4]. Studies showed that in these cases, no long-term motor deficits should be expected [4,8,18,19]. This might be why the two patients without the D-wave warning, or reversible D-wave warning, did not have new postoperative motor deficits. In two patients, there was an irreversible D-wave warning (patients 7 and 9, Table 4). In patient 7 from Table 4, D-waves were recorded during a kyphosis correction. However, D-wave recordings during scoliosis surgery appeared to produce false positive warnings and might therefore not be reliable for monitoring scoliosis patients [20]. In a more recent study, the authors reported that there might be added value in performing D-waves in patients undergoing thoracic spinal osteotomy or scoliosis correction, since it lowered the number of false positive outcomes of the mTc-MEPs [21]. However, they still reported false positive warnings of the D-wave monitoring. In patient 9 from Table 4, there was an irreversible mTc-MEP and D-wave warning. Although we do not completely understand the underlying physiology of why the D-wave deteriorated, the surgery lasted very long (10 h and 16 min) and the D-wave declined gradually over time. The final patient (patient 6, Table 4) with an irreversible mTc-MEP warning without new postoperative motor deficits, had what we consider a false positive warning.

### Limitations

Although this is the first extensive study investigating the reliability of two recording electrode types for detecting mTc-MEP warnings, it has its limitations. Even though this was a prospective study, for the secondary objective, the neurological outcomes of the patients were collected retrospectively. Because of this, not all MRC grades were reported, specifically in patients who underwent orthopaedic surgery or vascular surgery. However, it was not stated if there were new postoperative deficits or not. Moreover, pre- and postoperative neurological examinations were not always performed by the same physician. Secondly, the study’s results apply only to the specific surface and subcutaneous needle electrodes used in this study. Moreover, whether the use of surface electrodes is non-inferior to the use of subcutaneous needle electrodes for detecting mTc-MEP warnings in other muscles besides the TA muscles has yet to be investigated. Lastly, for lower motor neuron monitoring, it is probably advisable to record mTc-MEPs with intramuscular needles to be able to detect myotonic discharges.

## 5. Conclusions

The use of surface electrodes was non-inferior to the use of subcutaneous needle electrodes for detecting mTc-MEP warnings over the TA muscles during spinal cord monitoring. Moreover, reversible warnings never resulted in permanent new motor deficits of the left and right TA. Irreversible warnings resulted in a range from 50.0% (TA left) to 63.6% (TA right) in transient or permanent new postoperative motor deficits.

## Figures and Tables

**Figure 1 jcm-12-01753-f001:**
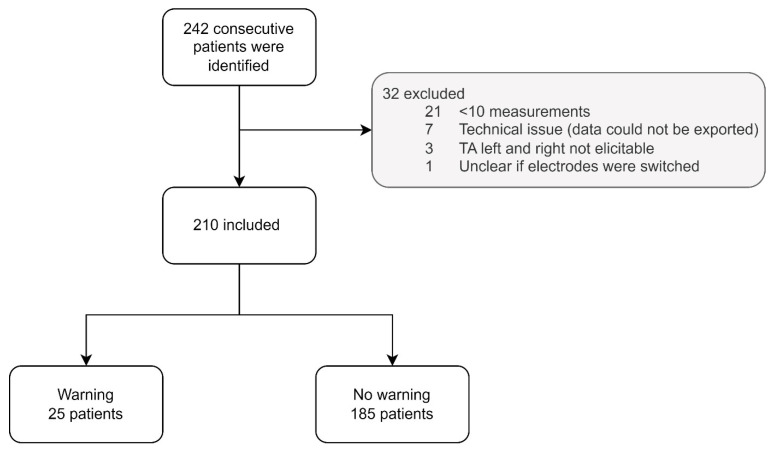
Flowchart patient selection. TA, tibialis anterior muscle.

**Figure 2 jcm-12-01753-f002:**
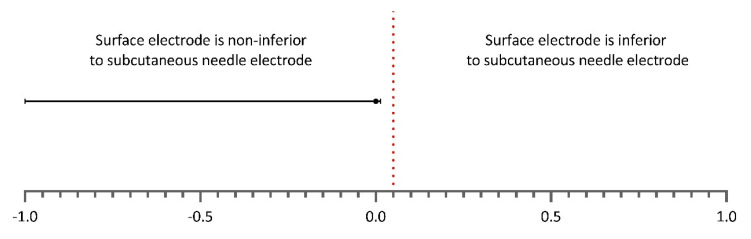
Difference in proportions of warnings detected by the use of subcutaneous needle and surface electrodes. The one-sided 95% CI is reported. The vertical dotted red line indicates the non-inferiority margin of 5.0%. As can be observed, the estimate and one-sided 95% CI remain on the left side of the dotted line, which implies non-inferiority of the use of surface recording electrode in detecting mTc-MEP warnings over the TA muscles.

**Figure 3 jcm-12-01753-f003:**
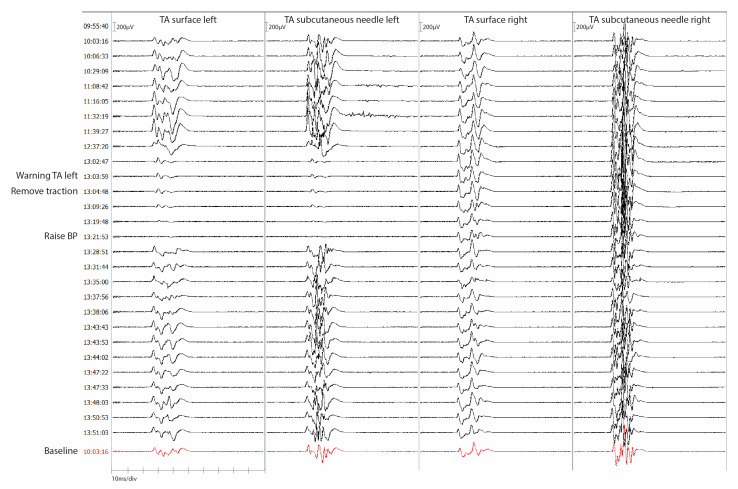
Case example of a reversible mTc-MEP warning for both surface and subcutaneous needle recording electrodes. A case example of patient 19 of Table 4. mTc-MEPs are recorded for the left and right TA with both surface and subcutaneous needle recording electrodes. The 15-year-old patient underwent scoliosis correction surgery by an anterior retroperitoneal approach due to a congenital hemivertebrae of S1. Preoperative left and right TA strength was intact (MRC 5). The left TA amplitudes disappeared at the same time for both recording electrodes. A warning was communicated to the surgeon. Traction on the frame was removed, and the blood pressure was raised after which the mTc-MEP amplitudes returned to normal. The patient woke up with a transient paresis (MRC 4) of the left TA which improved in a couple of days to an MRC 5. TA, tibialis anterior; BP, blood pressure.

**Table 1 jcm-12-01753-t001:** mTc-MEP warning criteria.

Warning Criteria mTc-MEP Amplitude	
Scoliosis surgery Neurosurgical surgeries Vascular surgeries	≥80% decrease in mTc-MEP amplitude ≥50% decrease in mTc-MEP amplitude ≥50% decrease in mTc-MEP amplitude

mTc-MEP, muscle-recorded transcranial electrical stimulation motor evoked potential.

**Table 2 jcm-12-01753-t002:** Patient characteristics.

Patient Characteristics	Patients (*n* = 210)	
**Median age at surgery in years (IQR)**	19 (15, 51)	
**Female N (%)**	135 (64.3)	
**Diagnosis N (%)**		
Orthopaedic surgery	123 (58.6)	
Idiopathic scoliosis	83 (39.5)	
Congenital scoliosis	3 (1.4)	
Neuromuscular scoliosis	19 (9.0)	
Syndromic scoliosis	15 (7.1)	
Kyphosis	3 (1.4)	
Neurosurgical surgery	67 (31.9)	
Extradural spinal tumour	2 (1.0)	
Intradural extramedullary tumour	22 (10.5)	
Intradural intramedullary tumour	15 (7.1)	
Intradural cauda equina tumour	9 (4.3)	
Tethered spinal cord	3 (1.4)	
ATSCH	4 (1.9)	
Thoracic HNP	6 (2.9)	
Transdural approach	3 (1.4)	
Extradural approach	3 (1.4)	
Cervical HNP (extradural approach)	1 (0.5)	
Extradural spinal nerve root tumour	3 (1.4)	
Transdural approach	2 (1.0)	
Extradural approach	1 (0.5)	
Degenerative spine instability (extradural approach)	1 (0.5)	
Trauma thoracic spinal cord compression (extradural approach)	1 (0.5)	
Endovascular thoracic/abdominal aortic aneurysm repair	20 (9.5)	
Mean surgery time in minutes (SD)	334 (111)	
**Elicitability N patients (%)**		
TA subcutaneous needle left	206 (98.1)	
TA surface left	206 (98.1)	
TA subcutaneous needle right	208 (99.0)	
TA surface right	208 (99.0)	
**Preoperative TA strength N (%)**	**Left**	**Right**
MRC 0	1 (0.5)	0 (0.0)
MRC 1	1 (0.5)	0 (0.0)
MRC 2	2 (1.0)	2 (1.0)
MRC 3	1 (0.5)	4 (1.9)
MRC 4	8 (3.8)	6 (2.9)
MRC 5	187 (89.0)	187 (89.0)
MRC < 5 *	10 (4.8)	11 (5.2)
**Postoperative TA strength N (%) ****	**Left**	**Right**
MRC 0	2 (1.0)	1 (0.5)
MRC 1	1 (0.5)	0 (0.0)
MRC 2	1 (0.5)	3 (1.4)
MRC 3	3 (1.4)	4 (1.9)
MRC 4	11 (5.3)	11 (5.3)
MRC 5	183 (87.6)	182 (87.1)
MRC < 5 *	8 (3.8)	8 (3.8)

* MRC grade was below five but not further specified in the medical reports. ** *n* = 209; in one patient, there was no postoperative neurological examination. ATSCH, anterior thoracic spinal cord herniation; HNP, hernia nucleus pulposus; MRC, medical research council scale; TA, tibialis anterior.

**Table 3 jcm-12-01753-t003:** The number of patients with an mTc-MEP warning in the left and/or right TA for both surface and subcutaneous needle recording electrodes.

mTc-MEP Warnings Tibialis Anterior Muscles (*n* = 210)			
		*Subcutaneous needle electrode*	
		+warning	−no warning
*Surface electrode*	+warning	25	0
	−no warning	0	185

mTc-MEP, muscle-recorded transcranial electrical stimulation motor evoked potential.

**Table 4 jcm-12-01753-t004:** Patient characteristics of the 25 patients that had a warning for the left and/or right TA.

Patient	Age	Sex	Diagnosis	mTc-MEP Warning: TAL, TAR or Both	Subsequent Action	Reversible	MRC TAL Pre-op	MRC TAL Post-op	MRC TAR Pre-op	MRC TAR Post-op	New Postoperative Deficits	D-Wave Measured	D-Wave Warning	D-Wave Reversible
1	57	F	Intradural Meningioma Th6	Both	None	Yes	5	5	5	5	No	Yes	No	
2	12	F	Syndromic scoliosis: SMs	Both	Autologous blood transfusion, decrease of NE concentration *	Yes	5	5	5	5	No	No		
3	52	M	HNP C6-7 (extradural)	Both	None (warning after small dura laceration, spontaneous recovery)	Yes	5	5	5	5	No	No		
4	85	M	Endovascular aneurysm repair	Both	Removal femoral artery sheaths	Yes	5	5	5	5	No	No		
5	17	F	Idiopathic scoliosis	Both	Increase BP, decrease propofol	Yes	5	5	5	5	No	No		
6	13	F	Aneurysmal bone cyst C5-C6	Both	None	No	5	5	5	5	No	No		
7	33	M	NF1, kyphosis	Both	Increase BP, decrease propofol concentration	No	5	5	5	5	No	Yes	Yes	No
8	29	F	ATSCH	Both	None	No	5	5	5	5	No	Yes	No	
9	40	F	ATSCH	Both	None	No	5	5	5	5	No	Yes	Yes	No
10	30	F	ATSCH	TAL	None	Yes	5	5	5	5	No	Yes	No	
11	68	F	Endovascular aneurysm repair	TAL	Removal femoral artery sheath	Yes	5	5	5	5	No	No		
12	13	M	Neuromuscular scoliosis: SMA	TAR	Removal hooks, decrease traction	Yes	<5	<5	<5	<5	No	No		
13	16	M	Neuromuscular scoliosis: CP	TAR	Removal rod	Yes	<5	<5	<5	<5	No	No		
14	74	M	Endovascular aneurysm repair	TAR	Removal femoral artery sheath	Yes	5	5	5	5	No	No		
15	72	F	Endovascular aneurysm repair	TAR	Removal femoral artery sheath	Yes	5	5	5	5	No	No		
16	63	M	Intramedullary lipoma Th1-4	Both	None	TAL yes, TAR no	5	5	5	4	Transient	Yes ***		
17	73	F	Spinal ependymoma C2-Th2	Both	Increase BP	No	5	3	5	0	Transient	Yes ****		
18	50	F	Ependymoma Th8-Th10	Both	None	No	5	4	5	4	Transient	Yes	No	
19	15	F	Congenital scoliosis	TAL	None (warning after mechanical replacement of a nerve root)	Yes	5	4	5	5	Transient	No		
20	65	F	HNP Th11-12 (extradural)	TAR (TAL not elicitable)	Increase BP	No	5	5	5	4	Transient	No		
21	20	M	Congenital scoliosis	Both	Increase BP, removal temporary rod	No	5	4	5	4	Transient	No		
22	54	F	HNP Th10-11 (transdural)	Both	Increase BP	No	5	3	5	0	TAL transient, TAR Permanent	No		
23	55	M	Chondrosarcoma th7-9	Both	None	No	5	4	5	4	Permanent	Yes	Yes	No
24	48	F	HNP Th8-9 (transdural)	TAL (TAR not elicitable)	Increase BP, irrigation spinal cord	No	5	5	5	2	TAL No, TAR Permanent	Yes	Yes	Yes
25	12	F	Neuromuscular scoliosis: SMA	Both	Decrease traction	TAL yes, TAR no	NA	3	NA	3	NA	No		

* Noradrenaline concentration was decreased to address possible ischemic effects due to vasoconstriction. *** D-wave amplitudes were too small and polyphasic for reliable D-wave monitoring. **** Muscle artifacts were considered too large for reliable D-wave monitoring. No muscle relaxants were administered due to the large asymmetry between the mTc-MEPs of the left and right lower limbs. mTc-MEP, muscle-recorded transcranial electrical stimulation motor evoked potential; TAL, left tibialis anterior muscle; TAR, right tibialis anterior muscle right; MRC, Medical Research Council Scale; ATSCH, anterior thoracic spinal cord herniation; NF1, neurofibromatosis type 1; HNP, herniated nucleus pulposus; CP, cerebral palsy; SMs, Smith Magenis syndrome; BP, blood pressure; NE, norepinephrine; NA, not available; SMA, spinal muscular atrophy.

**Table 5 jcm-12-01753-t005:** Number of patients with an mTc-MEP warning in the left TA and right TA separately for both surface and subcutaneous needle recording electrodes.

Tibialis Anterior Left (*n* = 204) *				
		New postoperative motor deficit		
		Permanent	Transient	None
mTc-MEP result	Warning: complete loss	1	2	1
	Warning: irreversible deterioration	0	2	4
	Reversible deterioration or loss	0	1	8
	No warning	0	0	185
**Tibialis Anterior Right (*n* = 206) ****				
		New postoperative motor deficit		
		Permanent	Transient	None
mTc-MEP result	Warning: complete loss	2	4	1
	Warning: irreversible deterioration	0	1	3
	Reversible deterioration or loss	0	0	9
	No warning	0	1	185

* In four patients, the tibialis anterior left was not elicitable. In one patient there was a warning; however, no preoperative neurological examination was performed. In one patient, there was no postoperative neurological examination. These six patients were excluded from this table. ** In two patients, the tibialis anterior right was not elicitable. In one patient, there was a warning; however no preoperative neurological examination was performed. In one patient, there was no postoperative neurological examination. These four patients were excluded from this table.

**Table 6 jcm-12-01753-t006:** Predictive values for the postoperative neurological outcome for the left TA and right TA separately.

Tibialis Anterior Left *				
	Predictive values of mTc-MEP warnings for new postoperative motor deficits % (CI)			
	Without RW (*n* = 195)	RW as TP (*n* = 204)	RW as FP (*n* = 204)	RW after application of causality guidelines (*n* = 204) ***
Sensitivity	100.0 (47.8–100.0)	100.0 (76.8–100.0)	100 (47.8–100.0)	100 (59.0–100.0)
Specificity	97.4 (94.0–99.1)	97.37 (94.0–99.1)	93.0 (88.5–96.1)	93.9 (89.6–96.8)
Positive predictive value	50.0 (29.6–70.4)	73.7 (54.1–86.9)	26.3 (17.7–37.2)	36.8 (25.2–50.2)
Negative predictive value	100.0 (100.0–100.0)	100.0 (100.0–100.0)	100.0 (100.0–100.0)	100.0 (100.0–100.0)
**Tibialis Anterior Right ****				
	Predictive values of mTc-MEP warnings for new postoperative motor deficits % (CI)			
	Without RW (*n* = 197)	RW as TP (*n* = 206)	RW as FP (*n* = 206)	RW after application of causality guidelines (*n* = 206) ***
Sensitivity	87.5 (47.4–99.7)	94.1 (71.3–99.9)	87.5 (47.4–99.7)	92.3 (64.0–99.8)
Specificity	97.9 (94.7–99.4)	97.9 (94.7–99.4)	93.4 (89.0–96.5)	95.9 (92.0–98.2)
Positive predictive value	63.6 (39.1–82.7)	80.0 (60.1–91.4)	35.0 (23.0–49.2)	60.0 (42.8–75.1)
Negative predictive value	99.5 (96.7–99.9)	99.5 (96.5–99.2)	99.5 (88.9–96.2)	99.5 (91.9–98.0)

* In four patients, the tibialis anterior left was not elicitable. In one patient there was a warning, however no preoperative neurological examination was performed. In one patient, there was no postoperative neurological examination. These six patients were excluded from this table. ** In two patients, the tibialis anterior left was not elicitable. In one patient there was a warning; however, no preoperative neurological examination was performed. In one patient, there was no postoperative neurological examination. These four patients were excluded from this table. *** For the left TA, two out of the nine patients with a reversible warning were considered true positive. For the right TA, five out of the nine patients with a reversible warning were considered true positive. RW, reversible warnings; TP, true positives; FP, false positives.

## Data Availability

The data may be available at a reasonable request.

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
