# Peer review of "Use of NEedle Versus suRFACE Recording Electrodes for Detection of Intraoperative Motor Warnings: A Non-Inferiority Trial. The NERFACE Study Part II"

_jcm, 2023, doi:10.3390/jcm12051753_

Round 1

Reviewer 1 Report

Dear Sirs,

Thank you for inviting me to review the paper entitled: ”Use of NEedle Versus suRFACE Recording Electrodes for 2 Detection of Intraoperative Motor Warnings: A Non-inferiority 3 Trial. The NERFACE Study Part II” - submitted to Journal of Clinical Medicine.

The authors pursued their research begun in NERFACE study part I. In the study the authors analyzed superiority and inferiority of needle subcutaneous versus surface electrodes for detection of intraoperative motor warnings.

The manuscript is interestingly, well written.

In table 2 the authors divided types of surgery by the surgical team. It would be clear to bold the services. It is not clear why subsection: intardural cauda equine tumor was extracted from the group (beside spinal nerve tumor or intradural tumor). The authors created subgroups for transdural or extradural removal of thoracic HNP without indicating type of approach (the same for cervical HNP)? The authors divided spinal nerve root tumor to transdural and extradural without giving information regarding the level of pathology or type of approach? The authors reported one patient with degenerative spine instability without revealing the level of pathology or type of approach. The same, the type of thoracic spine injury was not defined? What types of aneurysm were endovasculary treated?  

Kind regards,

Reviewer 2 Report

This paper compared needle and surface electrodes in the context of intraoperative monitoring. The result was that both types of electrodes performed equally well as regards detecting amplitude reductions. The authors added a second objective investigating how well warnings predicted postoperative injury.

The study is overall described well, although the level of detail can be considered excessive. As the results are straightforward, with prefect agreement between the two methods, the reader may well wonder why there are 16 pages. The paper can be shortened considerably without losing quality.

In fact, the room freed this way may be used by adding a subject that to my surprise was missing: why was the study performed? Putative reasons to carry out the study might firstly have been that the differing signal characteristics of the recorded potentials (as illustrated by Figure 3) would result in different diagnostic characteristics, or secondly that the methods different in costs, risks, stability of the signal, etc. Related to this is what readers should do with the results, or does it not matter?

The authors state that that a warning was defined as an amplitude reduction that was communicated to the surgeon, but do not make it clear whether they chose needle or surface electrodes, or both, for this purpose. In view of the results this is probably not relevant, but how was the study designed? Beforehand, it was unknown whether or not a method yielded more or fewer false positive or false negative results, so the design must have included a strategy to detect this and yet not harm patients. What was that strategy?

There is no information regarding signal analysis. Being unfamiliar with the practice of IONM, this reviewer wonders how amplitudes were measured, given that the signals seemed more suitable for an  analysis of area than of amplitude. The two types of signal obviously differ (Figure 3) in frequency content and intertrial variability. Does that affect the consequences of the study?

There are some minor matters:

- 'IONM aims to prevent spinal cord injury'; not true for all uses of IONM.

- The authors mention a part I of the NERFACE study (line 66), without providing a reference. Please add the reference or supply data.

- On line 115, how can you use a 50% as well as an 80% threshold?   

- The repeated use of 'TA left' and 'TA right' distracted from the content of the paper; please use  'left TA' and  'right TA'.
